# Anti-Virulence Potential and In Vivo Toxicity of *Persicaria maculosa* and *Bistorta officinalis* Extracts

**DOI:** 10.3390/molecules25081811

**Published:** 2020-04-15

**Authors:** Marina Jovanović, Ivana Morić, Biljana Nikolić, Aleksandar Pavić, Emilija Svirčev, Lidija Šenerović, Dragana Mitić-Ćulafić

**Affiliations:** 1Faculty of Biology, University of Belgrade, Studentski trg 16, 11158 Belgrade, Serbia; biljanan@bio.bg.ac.rs (B.N.); mdragana@bio.bg.ac.rs (D.M.-Ć.); 2Institute of General and Physical Chemistry, Studentski trg 12/V, 11158 Belgrade, Serbia; 3Institute of Molecular Genetics and Genetic Engineering, University of Belgrade, Vojvode Stepe 444a, 11042 Belgrade, Serbia; ivanamoric@imgge.bg.ac.rs (I.M.); sasapavic@imgge.bg.ac.rs (A.P.); seneroviclidija@imgge.bg.ac.rs (L.Š.); 4Faculty of Science, University of Novi Sad, Dositeja Obradovića 2, 21000 Novi Sad, Serbia; emilija.svircev@dh.uns.ac.rs

**Keywords:** *Persicaria maculosa*, *Bistorta officinalis*, antibiofilm activity, anti-quorum sensing activity, zebrafish

## Abstract

Many traditional remedies represent potential candidates for integration with modern medical practice, but credible data on their activities are often scarce. For the first time, the anti-virulence potential and the safety for human use of the ethanol extracts of two medicinal plants, *Persicaria maculosa* (PEM) and *Bistorta officinalis* (BIO), have been addressed. Ethanol extracts of both plants exhibited anti-virulence activity against the medically important opportunistic pathogen *Pseudomonas aeruginosa*. At the subinhibitory concentration of 50 µg/mL, the extracts demonstrated a maximal inhibitory effect (approx. 50%) against biofilm formation, the highest reduction of pyocyanin production (47% for PEM and 59% for BIO) and completely halted the swarming motility of *P. aeruginosa*. Both extracts demonstrated better anti-quorum sensing and antibiofilm activities, and a better ability to interfere with LasR receptor, than the tested dominant extracts’ constituents. The bioactive concentrations of the extracts were not toxic in the zebrafish model system. This study represents an initial step towards the integration of *P. maculosa* and *B. officinalis* for use in the treatment of *Pseudomonas* infections.

## 1. Introduction

One of the objectives of the United Nations Sustainable Development Goals is the achievement of good health and well-being for everyone, and the World Health Organization (WHO) is actively participating in supporting global efforts to integrate the best of traditional and complementary medicine with conventional medicine [1]. Unfortunately, traditional medicine faces a challenge from the lack of credible data; therefore, a comprehensive approach in the assessment of the efficacy and safety of traditionally used remedies is required.

Amongst the unique healthcare challenges we are facing in the 21st century, bacterial resistance to available therapeutics is of critical importance. Less than a century after the introduction of antibiotics into clinical practice, the number of antibiotic-resistant bacteria has significantly increased and still rises, making bacterial infections once again a substantial threat to humanity [2]. Furthermore, currently the predominant bacterial lifestyle is considered to be a biofilm form rather than a planktonic one. In biofilms, defined as multicellular communities immersed in a self-produced polymeric matrix, bacterial cells can be 10 to 1000 times less susceptible to antibacterial agents compared with their free-living forms [3]. Therefore, not only are new antibacterial agents urgently needed, but so are novel therapeutic strategies to fight bacterial infections.

Quorum sensing (QS), a bacterial cell-to-cell communication system, is involved in biofilm formation and the production of numerous virulence factors that are critical for bacterial pathogenicity. It relies on the synthesis, release, accumulation, sensing, and response to small diffusible signal molecules called autoinducers [4]. Interfering with QS signalling is expected to reduce virulence factor production, leading to less severe infections which could then be cleared by the host’s immune system. In addition, reducing biofilm formation may increase bacterial susceptibility to antibiotics and thus improve the effectiveness of antibacterial therapy. Importantly, since QS is not involved in bacterial growth, its inhibition does not impose a pressure on bacteria for resistance development, so the attenuation of virulence by interfering with QS presents a promising approach to combat infections [5].

Various plants belonging to the Polygonoideae subfamily (fam. Polygonaceae A.L. de Jussieu 1789) produce a large number of biologically important secondary metabolites, including flavonoids, anthraquinones, alkaloids and steroids which are often found in medicinal plants and are associated with health benefits [6]. Amongst them, the rhizome of *Bistorta officinalis* has been used in Chinese medicine to treat various inflammatory and infectious diseases, such as dysentery with bloody stools, diarrhea in acute gastroenteritis, carbuncles, scrofula and acute respiratory infection with cough [7]. On the other hand, the whole plant of *Persicaria maculosa* has been used in wound healing and against fungal infestations [8]. In addition, the roots of *B. officinalis* are used as a spice to flavor soups and stews, or in pastries when dried and ground [9], while the boiled leaves of *P. maculosa* have been used in the preparation of a very old traditional Italian dish known as pistic [10].

The aim of this study was to comprehensively assess the antibacterial and anti-virulence potential of ethanol extracts of *P. maculosa* (PEM) and *B. officinalis* (BIO), which are edible and medicinal, but also wild and widespread invasive plant species. The interference of the extracts and selected major compounds with the QS signalling pathways of Gram-negative bacteria was also examined. Their safety for human usage was addressed by the evaluation of their toxicity *in vivo* in the zebrafish (*Danio rerio*) model system.

## 2. Results and Discussion

### 2.1. Antibacterial Effect of the Extracts

According to the extensive literature, different species of the Polygonoideae subfamily are used in traditional medicine for the treatment of various health problems, but data confirming their potential to fight bacterial infections are both scarce and incomplete [11]. Although the antibacterial activities of different PEM and BIO extracts were assessed in a few recent studies, the extracts were typically tested in disk or well diffusion assays, often in concentrations too high (up to 30 mg/mL) to be relevant [12,13,14,15]. Therefore, we re-evaluated the activity of PEM and BIO extracts against several important human pathogens in a more quantitative way and demonstrated that the BIO extract has moderate antibacterial activity against *Staphylococcus aureus* with the minimum inhibitory (MIC) and minimum bactericidal (MBC) concentrations of 156 µg/mL and 312 µg/mL, respectively, but weak bactericidal activity against *Pseudomonas aeruginosa* PAO1 with MIC of 1 mg/mL. The extracts showed very weak activity against *Salmonella enterica* subspecies *enterica* serovar Enteritidis with MIC values of 2.5 mg/mL and 5 mg/mL for the BIO and PEM extracts, respectively. None of the extracts showed antibacterial activity against *Escherichia coli*, *Shigella flexneri*, *Listeria monocytogenes* or *Enterococcus faecalis* even at concentrations as high as 5 mg/mL.

Our results showed that the ethanol extracts of both plants have a moderate to no effect on planktonic cells of the tested bacterial strains, and prompted further research towards the examination of their anti-virulence potential, i.e., their activity against bacterial virulence machinery, which is required for host damage and disease development.

### 2.2. Antibiofilm Activity

The WHO [16] recently published a list of bacteria that pose the greatest threat to human health, in order to help in prioritizing research. Thus, to evaluate the potential of the extracts to prevent biofilm formation, both Gram-negative and Gram-positive representative bacterial strains were selected and the extracts’ activities were tested at their subinhibitory concentrations. For the PEM and BIO antibiofilm activity study, we chose *P. aeruginosa,* which belongs to the “critical pathogen” category that comprises only Gram-negative bacteria, *S.* Enteritidis (Gram-negative), and *S. aureus* (Gram-positive) bacterial species, that are categorized by the WHO as pathogens of “high” priority.

Both plant extracts inhibited biofilm formation in *P. aeruginosa* PAO1 at concentrations of 50 µg/mL and 100 µg/mL (Figure 1A), but the effects were not dose-dependent. The PEM extract inhibited biofilm formation in *S*. Enteritidis by up to 50% at 100 µg/mL, while the BIO extract reduced biofilm formation by 60% at a concentration of 50 µg/mL (Figure 1B). The PEM and BIO extracts showed similar antibiofilm effects against Gram-negative bacterial species, but in *S. aureus*, in the presence of PEM, biofilm formation was stimulated in a dose-dependent manner, while BIO had the opposite effect—it prevented biofilm formation by up to 60% with an increased concentration (Figure 1C).

These results reflect the intrinsic differences between Gram-negative and Gram-positive bacteria, as well as imply the involvement of different molecular mechanism(s) of action, and emphasize the complexity of the effect(s) caused by plant extracts. As such, they prompted our research towards an in-depth investigation of the anti-QS activities of the extracts in Gram-negative bacteria.

### 2.3. Detection of Anti-QS Activity of the Extracts and Their Effects on Selected Virulence Factors in Gram-Negative Bacteria

Next, the anti-QS activity of the extracts was assessed by using a *Chromobacterium violaceum* CV026 assay. The production of the purple pigment violacein in the *C. violaceum* CV026 strain is induced with an exogenously provided acylhomoserine lactone, such as N-(hexanoyl)-L-homoserine lactone (HHL), which binds to its receptor, CviR, acting as a transcriptional activator. When the specific inhibitor competes with HHL and binds to the receptor, the synthesis of violacein is hindered [4]. In the presence of the extracts, especially in the case of BIO, opaque zones around the cellulose disks were observed (Figure 2A), indicating that one or more components present in the extracts could compete with HHL for CviR, and thus affected violacein production.

Further studies on the anti-QS activities of the extracts were performed in assays with *P. aeruginosa*, which is a medically important opportunistic pathogen, but also a model organism often used to study QS in Gram-negative bacteria. The interference of the PEM and BIO extracts with QS in *P. aeruginosa* was examined by testing their effects on pyocyanin synthesis, a blue-green pigment that promotes virulence through interference with several functions in host cell [17]. It has been shown that pyocyanin, a natural phenazine molecule, has numerous antagonistic effects, including pro-inflammatory and free radical effects that lead to cellular damage and death [18]. Both extracts affected pyocyanin production only at 50 µg/mL, inhibiting the pyocyanin level in the supernatant by 47% and 59% for PEM and BIO, respectively (Figure 2B). Both extracts exhibited a non-monotonic dose–response, where the highest tested concentration of BIO even increased pigment production by 43%.

The interference of the extracts with QS in *P. aeruginosa* was further addressed by testing their effects on swarming motility, which is a QS-regulated social phenomenon involving the rapid coordinated movement of bacteria over surfaces, and is related to an elevated adaptive resistance to antibiotics. Bacterial swarming motility is important for the invasion of tissues and their successful colonization [19,20]. Both extracts completely halted the swarming motility of *P. aeruginosa* (Figure 2C).

### 2.4. In Vitro Cytotoxicity Assessment

The cytotoxic potential of PEM and BIO was tested on MRC-5 cells and the obtained results showed only weak cytotoxicity with IC_50_ values of 1 mg/mL and 1.5 mg/mL for PEM and BIO, respectively.

### 2.5. Identification of Compounds in the Extracts

Extensive studies of the chemical properties of *P. maculosa* and *B. officinalis* have showed that their extracts are rich in various flavonoids and phenolic acids [7,8,9]. The phenolic contents of the ethanol extracts of PEM and BIO are given in Table 1. Twenty-six different phenolics—10 phenolic acids, eight flavonoids and eight flavonoid glycosides—have been identified in PEM extract, although many in concentrations below 0.5 mg per gram of dry weight. On the other hand, only eight phenolic compounds (five phenolic acids and three flavonoids) have been detected in BIO extract. The most abundant compounds in PEM extract were identified as quinic acid (QA), chlorogenic acid (Cha), gallic acid (GA), quercetin-3-*O*-glucoside (Quer-3-*O*-Glc), hyperoside and rutin, while in BIO extract, chlorogenic acid and catechin (Cat) were the dominant phenolics.

### 2.6. Identification of QS Signalling Pathways Affected by Extracts and Selected Phenolic Components

Prior to analysing the interference of the extracts and selected constituents with QS signalling pathways, we quantified the anti-QS and antibiofilm activity of five of the extracts’ components—QA, GA, ChA, Cat and Quer-3-*O*-Glc (Table 1). The inhibition of violacein production in *C. violaceum* CV026 with pure compounds was observed for all except GA (Figure 3A). When their effects on *P. aeruginosa* PAO1 biofilm development were tested, both stimulating and reducing activities were observed (Figure 3B). The strongest antibiofilm activities were evidenced for Quer-3-*O*-Glc (31%) and GA (28%), both at 50 µg/mL concentration, while QA exhibited only a stimulatory effect on biofilm formation regardless of the concentration. The obtained results were in accordance with published data [21,22,23,24].

The interference of the extracts and pure compounds with three major interconnected QS systems of *P. aeruginosa: las*, *rhl* and *pqs*, was next addressed. The numerous *P. aeruginosa* virulence factors, including pyocyanin synthesis and swarming motility, as well as the production of elastases, rhamnolipids, lipase, alkaline protease, exotoxin A and hemolysin, are under complex network regulation, mediated by these signalling pathways [25]. Three *P. aeruginosa* strains—PA14-R3, PAOJP2 and PAO1 Δ*pqsA—*were used as biosensors to detect and quantify the interaction of the tested substances with LasR, RhlR and PqsR receptors, respectively. The PEM and BIO extracts reduced the activity of the LasR receptor by 23% and 30%, respectively (Table 2). Similarly, Quer-3-*O*-Glc and Cat also interfered with the *las* signalling pathway and inhibited LasR activity by approximately 19% (Table 2). With the exception of ChA, which showed a slight stimulation of RhlR, all of the other tested substances did not affect RhlR activity. On the other hand, PqsR activity was generally stimulated, with the exception of QA and Quer-3-*O*-Glc, with no effect detected. The findings regarding the effects of pure compounds were similar to previously reported results [22,26].

Our results revealed that the extracts of *B. officinalis* and *P. maculosa* induced a higher effect than the tested pure compounds, which was especially evident in the case of BIO, thus indicating the synergistic activity of the extracts’ constituents. Both extracts exhibited moderate anti-QS activity through *las* signalling pathway inhibition, with Quer-3-*O*-Glc and Cat identified as the constituents contributing to the effect. Moreover, the evidenced stimulation of PqsR activity could explain the observed stimulation of pyocyanin production when the extracts were applied in higher concentrations (Figure 2B).

### 2.7. Toxicity Assessment in the Zebrafish Model

The zebrafish *(Danio rerio)* model was used to examine the in vivo toxicity and teratogenicity of the extracts and two pure compounds, Cat and Quer-3-*O*-Glc, which showed an inhibitory effect on the *las* signalling system (Table 2). In recent years, the zebrafish model emerged as a universal platform for the toxicity evaluation of plant extracts and bioactive molecules due to the high molecular, genetic, physiological and immunological similarity between zebrafish and humans [27], simplifying the path to clinical trials and reducing failure at later stages of testing [27,28]. To the best of our knowledge, none of the extracts of the tested plants have ever been examined in the zebrafish model to date. The zebrafish embryos were exposed to different concentrations of both the extracts and pure compounds over a period of five days and evaluated for survival and the appearance of cardiotoxic, hepatotoxic and teratogenic malformations, presented in Table 3. The obtained data show that both of the extracts, Quer-3-*O*-Glc and Cat had no adverse effect on the embryos’ survival and development at concentrations up to 125 µg/mL, 30 µg/mL and 50 µg/mL, respectively (Figure 4A). Furthermore, no cardiotoxic response was achieved in zebrafish embryos exposed to BIO and PEM applied in doses of 200 µg/mL and 250 µg/mL, respectively (Figure 4B). According to the results, Cat was not toxic in the applied concentration range (up to 50 µg/mL). The toxicity of the other tested substances ranged as follows: PEM (LC_50_ = EC_50_ = 247.2 µg/mL) < BIO (LC_50_ = 207.7 µg/mL, EC_50_ = 184.6 µg/mL) < Quer-3-*O*-Glc (LC_50_ = 51.4 µg/mL, EC_50_ = 38.7 µg/mL) (Figure 4A). In addition, BIO (250 µg/mL) and Quer-3-*O*-Glc (50 µg/mL) induced numerous teratogenic malformations, as presented in Figure 4B.

In summary, the PEM and BIO extracts exhibited anti-QS activity in *P. aeruginosa*, a medically important pathogen, at concentrations which were multifold lower than their IC_50_ and LC_50_ values, thus they can be regarded as safe. In addition, for the first time it has been shown that catechin, a dietary flavonoid frequently present in edible plants and beverages, at the tested concentration, is also safe for human use.

## 3. Materials and Methods

### 3.1. Chemicals and Media

Brain heart infusion (BHI) medium/broth was purchased from Lab M (Heywood, UK), while Luria broth/agar (LB/LA) and Mueller–Hinton broth (MHB) were provided by HiMedia (Mumbai, India). TetraMin^TM^ flakes were provided by Tetra Melle (Melle, Germany). Materials purchased from Sigma-Aldrich (St. Louis, MO, USA) were: resazurin, HHL, 2-heptyl-4-quinolone (HHQ), N-3-oxo-dodecanoyl-L-homoserine lactone (3-oxo-C12-HSL), N-butyryl-L-homoserine lactone (C4-HSL), QA, GA, ChA, Cat, Quer-3-*O*-Glc, Dulbecco’s modified Eagle’s medium (DMEM), fetal bovine serum (FBS), penicillin–streptomycin mixture, phosphate-buffered saline (PBS), trypsin from porcine pancreas, trypan blue, DMSO, 3-(4,5-dimethylthiazol-2-yl)-2,5-diphenyl tetrazolium bromide (MTT), reference standards of phenolic compounds and tricaine solution. HPLC gradient grade methanol was purchased from J. T. Baker (Deventer, The Netherlands) and p.a. formic acid from Merck (Darmstadt, Germany). All other chemicals (molecular biology grade) were purchased from local companies.

### 3.2. Plant Material, Extract Preparation and Chemical Analysis

The aerial parts of *P. maculosa* and the rhizome of *B. officinalis* were collected on the mountains Tara (western Serbia) and Stara Planina (eastern Serbia), respectively. The voucher specimens were deposited at the herbarium of the Department of Biology and Ecology, Faculty of Natural Sciences, University of Novi Sad, Serbia (voucher numbers for *P. maculosa* and *B. officinalis* are 2-1689 and 2-1674, respectively).

From the dried plant material, 5–15 g was extracted by maceration with 80% ethanol (10 mL/g of drug) for 72 h at room temperature. The filtered extract of the aerial parts of *P. maculosa* was evaporated in a vacuum, suspended in water and purified by liquid–liquid extraction with petroleum ether. The petroleum ether layer was washed with methanol and the methanol extract was pooled with the aqueous layer. The filtered rhizome extract of *B. officinalis* did not require a petroleum ether purification step, since pigments like chlorophyll are not present in rhizomes. Subsequently, the extracts were evaporated to dryness in a vacuum evaporator (< 45 °C) and dissolved in DMSO to a final concentration of 200 mg/mL.

The phytochemical profiles of *P. maculosa* and *B. officinalis* were determined by quantitative LC-MS/MS analysis using an Agilent Technologies 1200 Series High-Performance Liquid Chromatograph coupled with an Agilent Technologies 6410A Triple-Quad Tandem Mass Spectrometer with an electrospray ion source, and controlled by Agilent Technologies MassHunter Workstation software (ver. B.03.01) [29].

### 3.3. Microbial Strains and Growth Conditions

The following bacterial strains were used: *E. coli* (ATCC 8739), *S. flexneri* (ATCC 9199), *S.* Enteritidis (ATCC 13076), *S. aureus* (ATCC 25923), *L. monocytogenes* (ATCC 19111), *E. faecalis* (ATCC 29212), *P. aeruginosa* PAO1 (DSMZ 22644) and *P. aeruginosa* PA14. Bacterial overnight cultures were prepared in MHB broth, except *L. monocytogenes* cultures, which were grown in BHI.

### 3.4. Antibacterial Susceptibility Test for Planktonic Cells

MIC and MBC values were determined according to the standard broth microdilution method recommended by the Clinical and Laboratory Standards Institute [30]. Briefly, the exponential cultures of the tested strains containing 10^8^ CFU/mL were pelleted and resuspended in 0.01M MgSO_4_ to achieve 10^6^ CFU/mL. The MIC assay was performed in 96-well microtiter plates by making serial twofold dilutions of the tested substances in appropriate media. To each 200 µL of the dilutions, 20 µL of bacterial suspension were added. After incubation (18–20 h at 37 °C), an aqueous solution of resazurin was added to each well in a final concentration of 0.0675 mg/mL. Following incubation (up to 3 h at 37 °C), MIC values were determined as the lowest concentrations of the tested substances without a visible color change. MBC values were determined by plating 10 µL of the samples from wells without visible growth onto the appropriate solid medium. For each strain, three independent experiments were performed in triplicate.

### 3.5. Quorum Sensing Inhibition Assays

#### 3.5.1. Static Biofilm Formation Inhibition Assay

An assay was performed in a 96-well microtiter plate using a crystal violet (CV) staining method [31]. Briefly, LB medium containing either an extract or pure compound in concentrations of 10, 50 and 100 µg/mL, or 0.1% DMSO (negative control) was added into each well and inoculated with bacteria (1 × 10^6^ CFU/mL for *S. aureus* and *S.* Enteritidis, and 5 × 10^7^ CFU/mL for *P. aeruginosa* PAO1). After incubation for 24 h at 37 °C, the wells were thoroughly washed with sterile water and air-dried for 30 min. The formed biofilm was stained with 0.1% CV and incubated at room temperature for 15 min. The excess stain was removed and the plates were thoroughly washed. The dye bound to the biofilm was solubilized with 100 µL of 30% acetic acid. After 10 min of incubation, absorbance at 570 nm was measured using an Infinite^®^ 200 PRO (Tecan Group Ltd., Männedorf, Switzerland). The experiment was performed in sextuplicate and repeated three times.

#### 3.5.2. Chromobacterium violaceum Assay

A QS inhibition assay was performed on *C. violaceum* CV026 [32] as previously reported [4]. Violacein production was induced with 5 µM HHL. Bacteria were seeded in molten semi-solid LB agar (0.3%) that was poured over LA media. Cellulose disks containing an extract (250 µg/disk), pure compound (100 µg/disk) or DMSO (5 µL) were placed on solidified semi-solid agar and incubated for 20 h at 30 °C. The inhibition of violacein synthesis was detected by the presence of opaque zones on a purple background.

#### 3.5.3. Pyocyanin Production

A pyocyanin assay was performed on *P. aeruginosa* PA14 as reported previously [33]. The production of pyocyanin was quantified in overnight cultures growing in the presence of an extract (in concentrations of 10, 50 and 100 µg/mL) or 0.1% DMSO, using a UV–vis spectrophotometer Ultrospec 3300 pro (Amersham Biosciences, USA) at 695 nm. The production of pyocyanin was normalized per cell density (OD_600_). The results were presented relative to normalized pyocyanin production in the presence of 0.1% DMSO. The experiments were performed in triplicate and repeated three times.

#### 3.5.4. Swarming Assay

Swarm agar plates were made with M8 medium (50 mM Na_2_HPO_4_, 25 mM KH_2_PO_4_, 4 mM NaCl), supplemented with glucose (0.2%), casamino acid (0.5%), MgSO_4_ (1 mM) and solidified with agar (0.5%) in the presence of an extract (50 µg/mL) or DMSO (0.1%). The plates were inoculated with an exponential bacterial culture (3 μL, OD_600_ = 0.4–0.6) [34] and incubated at 37 °C for 24 h. The experiment was repeated twice in triplicate.

#### 3.5.5. Interference of Extracts and Components with QS Pathways

Overnight cultures of the biosensors *P. aeruginosa* PAO1 Δ*pqsA* (CTX *lux::pqsA*) [35], PAOJP2/pKD-rhlA (ΔrhlA P*rhlA::lux*) [17] and PA14-R3 (ΔlasI P*rsal::lux*) [36] were diluted to OD_600_ = 0.045 and incubated with the extracts or their major components in the presence of the specific autoinducers3oxo-C12HSL, C4-HSL and HHQ, respectively. Cell density and bioluminescence were measured after 4 h of incubation using an Infinite^®^ 200 PRO (Tecan Group Ltd., Switzerland). Bioluminescence was normalized per OD_600_. The assays were carried out three times in quadruplicate.

### 3.6. Cytotoxicity Assay

Normal human lung fibroblast cells MRC-5 (ECACC 84101801, UK) were cultured in DMEM supplemented with 10% FBS and 1% penicillin–streptomycin mixture at 37 °C. The cytotoxic effects of the plants’ extracts were assessed by an MTT assay as previously described [37]. Briefly, MRC-5 cells were seeded into 96-well plates at a density of 4 × 10^4^ cells/well and exposed to a series of twofold dilutions of the extracts (0.25–2 mg/mL) for 24 h. Following incubation, the medium was replaced with one containing MTT in a final concentration of 0.5 mg/mL, and the cells were incubated for 3 h. Then the medium was removed, the formazan crystals were dissolved in DMSO and the A_570_ was measured using an Infinite^®^ 200 PRO (Tecan Group Ltd., Menndorf, Switzerland). Three independent experiments were performed in sextuplicate.

### 3.7. In Vivo Toxicity Evaluation

The evaluation of the toxicity of the extracts and pure constituents was performed in the zebrafish (*Danio rerio*) model according to the general rules of the Organisation for Economic Co-operation and Development (OECD) Guidelines for the Testing of Chemicals [38], following a previously described procedure [39]. Embryos of wild-type zebrafish were raised to the adult stage in a temperature- and light-controlled zebrafish facility at 28 °C and standard 14:10 h light–dark photoperiod. They were regularly fed with TetraMin^TM^ flakes and brine shrimp twice a day and once a day, respectively. Zebrafish embryos were produced by pairwise mating, collected, distributed into 24-well plates containing 10 embryos per well and 1 mL embryo water (0.2 g/L of Instant Ocean^®^ Salt in distilled water) and raised at 28 °C. For assessing lethality, developmental toxicity and cardiotoxicity, the embryos staged at 6 h post-fertilization (hpf) were exposed to different concentrations of the extracts and pure compounds. The extracts and pure constituents were tested in concentrations ranging up to 250 µg/mL and 50 µg/mL, respectively. Embryo water was used as a negative control. The experiments were performed at least three times using 30 embryos per concentration.

The apical endpoints for the toxicity evaluation (Table 3) were recorded at 24, 48, 72, 96 and 120 hpf using an inverted microscope (CKX41; Olympus, Japan), as previously described [40]. Dead embryos were counted and discarded every 24 h. At 120 hpf, embryos were inspected for the heartbeat rate, anesthetized by the addition of 0.1% tricaine solution, photographed and killed by freezing at −20 °C for ≥ 24 h. A lethal dose causing the mortality of 50% of embryos (LC_50_) and a dose causing malformations in 50% embryos (EC_50_) were determined by the program ToxRatPro (ToxRat Solution GmbH, Germany, version 2.10.05), using a probit analysis with linear maximum likelihood regression.

### 3.8. Statistical Analysis

Data obtained from the static biofilm formation inhibition and pyocyanin production assays were analysed by an analysis of variance (one-way ANOVA, Dunnett’s multiple comparisons test) using GraphPad Prism software. Excel statistical software 2016 MSO (Student’s *t*-test) was used to determine the statistical significance of the MTT and QS inhibition results. The values in the figures are representative of three independent experiments performed in sextuplicate, presented as mean ± SD. DMSO (0.1%) was used as a negative control. The level of statistical significance was defined as p < 0.05 and has been presented in the figures with an asterisk (*).

## 4. Conclusions

The anti-virulence activity of the plant extracts of the Polygonoideae subfamily that are used in traditional medicine, has barely been investigated. To the best of our knowledge, for the first time the antibacterial, antibiofilm and anti-QS potential of extracts of *P. maculosa* and *B. officinalis* have been tested in a comprehensive manner. In addition, their embryotoxicity was addressed in order to evaluate whether they could be regarded as beneficial traditional remedies and safe for use, and as such present adequate candidates for further rigorous testing prior to their integration into modern medicine.

Our results have revealed that the ethanol extract of *B. officinalis* has moderate to weak antibacterial activity, while both extracts exhibited anti-virulence activity against the medically important opportunistic pathogen *P. aeruginosa* through the inhibition of the LasR receptor. Moreover, both extracts performed better as anti-virulent agents than their predominant pure constituents. Demonstrating the extracts’ safety, we have revealed that these common and widely distributed weedy plants have the potential to become a part of conventional medical practice for the treatment of bacterial infections. However, this study is just an initial step, and further testing is needed, starting with the evaluation of the extracts’ activities against clinical isolates of *P. aeruginosa,* followed by an unambiguous confirmation of the safety of these extracts to fight bacterial infections. In accordance with the WHO guidelines, additional pharmacological, observational and clinical studies should be carried out in order to satisfy the criteria of conventional medical drug use.

## Figures and Tables

**Figure 1 molecules-25-01811-f001:**
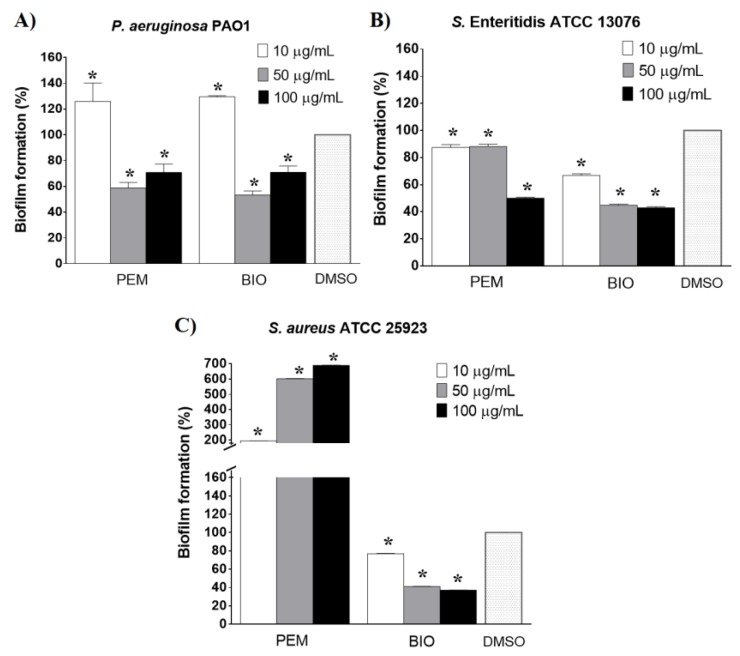
Effects of *P. maculosa* (PEM) and *B. officinalis* (BIO) extracts on biofilm formation of *P. aeruginosa* PAO1 (**A**), *S.* Enteritidis ATCC 13076 (**B**) and *S. aureus* ATCC 25923 (**C**).

**Figure 2 molecules-25-01811-f002:**
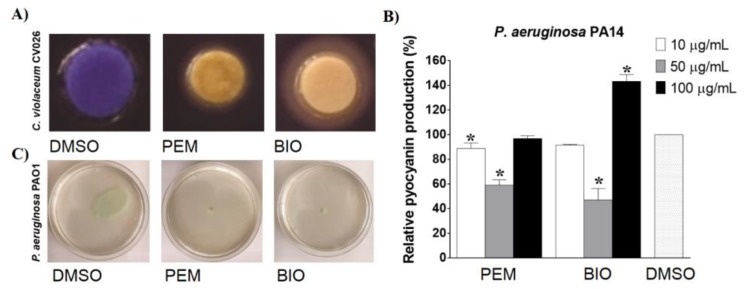
Anti-quorum sensing activity of the PEM and BIO extracts. Effect of the extracts on (**A**) violacein production detected in *C. violaceum* CV026 disk assay (250 μg/disk), (**B**) pyocyanin synthesis in *P. aeruginosa* PA14 and (**C**) swarming motility in *P. aeruginosa* PAO1 (50 µg/mL).

**Figure 3 molecules-25-01811-f003:**
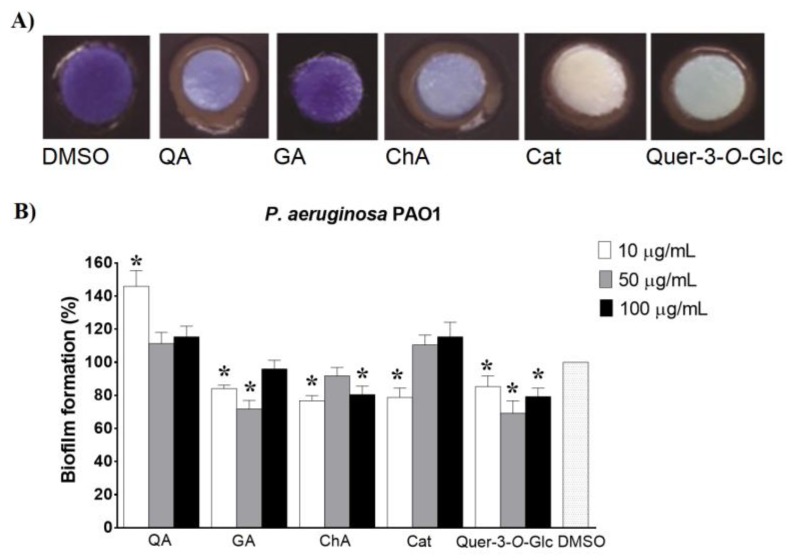
Anti-QS and antibiofilm activity of pure compounds. Effect of pure compounds on (**A**) violacein production detected in *C. violaceum* CV026 disk assay (100 μg/disk), and (**B**) biofilm formation in *P. aeruginosa* PAO1 (10, 50 and 100 µg/mL).

**Figure 4 molecules-25-01811-f004:**
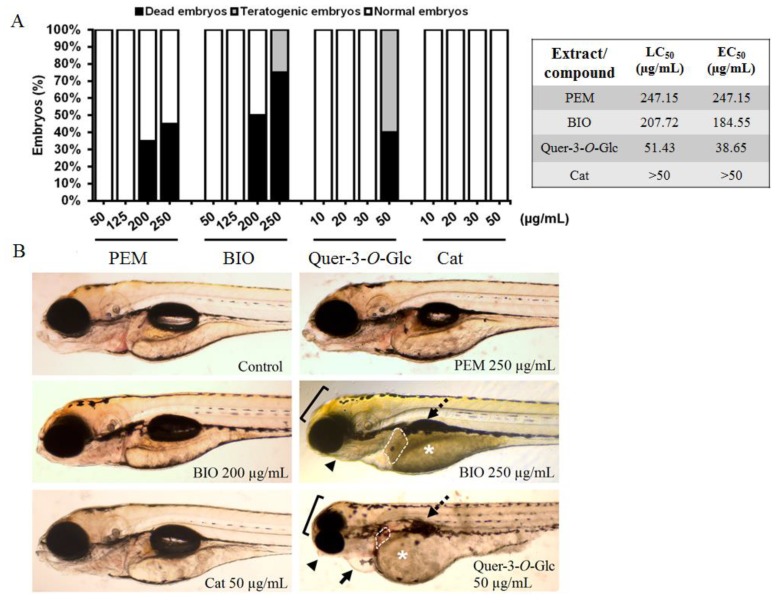
Toxicity assessment of PEM, BIO, Cat and Quer-3-*O*-Glc LC_50_ values (μg/mL). (**A**) Morphology of zebrafish embryos after different treatments and (**B**) Teratogenic malformations induced with BIO (250 µg/mL) and Quer-3-*O*-Glc (50 µg/mL): malformed head and eyes (bracket), malformed jaw (arrowhead), pericardial oedema (arrow), no inflated swim bladder (dashed arrow) and the signs of hepatotoxicity, such as dark yolk (asterisk) and dark liver (dashed area).

**Table 1 molecules-25-01811-t001:** Composition of *P. maculosa* (PEM; herb) and *B. officinalis* (BIO; rhizome) ethanol extracts.

Compound	Content (mg/g dw) ^a^	
	PEM	BIO
***Phenolic acids***		
*Quinic acid*	12.8 ± 1.28	1.14 ± 0.11
*Gallic acid*	2.47 ± 0.22	0.97 ± 1.69
*Chlorogenic acid*	6.41 ± 0.32	33.89 ± 1.69
Protocatechuic acid	0.21 ± 0.02	0.01 ± 0.00
2,5-dihydroxybenzoic acid	0.01 ± 0.00	Nd
p-hydroxybenzoic acid	0.04 ± 0.00	Nd
Caffeic acid	0.02 ± 0.00	0.02 ± 0.00
Syringic acid	0.01 ± 0.00	Nd
Coumaric acid	0.06 ± 0.01	Nd
Ferulic acid	0.05 ± 0.01	Nd
***Flavonoids***		
*Catechin*	0.19 ± 0.02	14.92 ± 1.49
Epicatechin	0.15 ± 0.02	1.67 ± 0.17
Hyperoside	4.93 ± 0.30	Nd
Rutin	2.65 ± 0.08	Nd
*Quercetin-3-O-glucoside*	11.71 ± 0.35	Nd
Quercetin-3-*O*-L-rhamnoside	2.10 ± 0.13	Nd
Kaempferol-3-*O*-glucoside	1.44 ± 0.06	Nd
Epigallocatechin gallate	0.30 ± 0.09	Nd
Vitexin	0.03 ± 0.00	Nd
Apigenin-7-*O*-glucoside	0.22 ± 0.01	Nd
Myricetin	0.02 ± 0.01	Nd
Luteolin-7-*O*-glucoside	0.14 ± 0.00	Nd
Quercetin	0.27 ± 0.08	0.01 ± 0.00
Naringenin	0.08 ± 0.01	Nd
Luteolin	0.13 ± 0.01	Nd
Apigenin	0.56 ± 0.04	Nd

^a^ Results are given as the concentration (mg per g of dry weight) ± relative standard deviation. In bold and italic font are the labelled compounds selected for further testing; Nd—not detected.

**Table 2 molecules-25-01811-t002:** Effects of the extracts PEM and BIO, and their major components, on the activity of QS receptors detected by bioluminescence measurements.

Extract or Compound	Relative Receptor Activity (%) *		
	LasR	RhlR	PqsR
PEM	77 ± 1	88 ± 1	153 ± 4
BIO	70 ± 1	90 ± 2	120 ± 10
QA	120 ± 2	103 ± 4	108 ± 8
GA	92 ± 3	95 ± 3	150 ± 35
ChA	108 ± 2	119 ± 2	125 ± 5
Cat	82 ± 2	98 ± 6	116 ± 7
Quer-3-*O*-Glc	81 ± 4	93 ± 4	98 ± 10

***** Values are relative to dimethyl sulfoxide (DMSO)-treated samples and are presented as mean ± SD. Applied concentrations were 50 µg/mL for extracts and 100 µg/mL for pure constituents.

**Table 3 molecules-25-01811-t003:** Lethal and teratogenic effects monitored in the treated zebrafish (*Danio rerio*) embryos at different hours post-fertilization (hpf).

Category	Developmental Endpoints	Exposure Time (hpf)				
		24	48	72	96	120
Lethal effect	Coagulated egg ^a^	●				
	Tail not detachment	●				
	No somite formation	●				
	Lack of heart beat	●				
Teratogenic effect	Malformation of head		●	●	●	●
	Malformation of eyes ^b^		●	●	●	●
	Malformation of jaw		●	●	●	●
	Malformation of sacculi/otoliths ^c^		●	●	●	●
	Malformation of chorda		●	●	●	●
	Malformation of tail ^d^		●	●	●	●
	Scoliosis		●	●	●	●
	Yolk oedema		●	●	●	●
	Yolk deformation		●	●	●	●
	Growth retardation ^e^			●	●	●
	Hatching				●	●
Cardiotoxicity	Pericardial oedema		●	●	●	●
	Heart morphology				●	●
	Heart beating rate (beats/min)					●

^a^ No clear organ structure was recognized. ^b^ Abnormality in shape and size of eyes. ^c^ Presence of none, one or more than two otoliths per sacculus, as well as reduction and enlargement of otoliths and/or sacculi (otic vesicles). ^d^ Tail malformation was recorded when the tail was bent, twisted or shorter than in control embryos as assessed by optical comparison. ^e^ Growth retardation was recorded by comparisons with the control embryos’ body length (after hatching, at 72 hpf, and onwards) using an inverted microscope.

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
