# Peer review of "Anti-Virulence Potential and In Vivo Toxicity of Persicaria maculosa and Bistorta officinalis Extracts"

_molecules, 2020, doi:10.3390/molecules25081811_

Round 1
Reviewer 1 Report
The second version of molecules-765605 became much more qualitative and can be accepted for publication in Molecules.
Reviewer 2 Report
The manuscript reports an interesting evaluation of the effect of ethanol extracts of Persicaria maculosa and Bistorta officinalis as antimicrobial substances, principally focusing on the effect against Pseudomonas aeruginosa.
The manuscript is well written and the data sound interesting, particularly because the two species are widely distributed weedy plants. The research has included also aspects related to the cytotoxicity and is an in-depth study. For this reasons, the data are worthy to be better discussed.
Specific comments:
Introduction:
L36 “is actively participating” instead of “participate”
Results and discussion:
It is not clear why, after a first screening, the authors decided to concentrate their efforts on few strains and at the end only on P. aeruginosa. Which was the reason of this decision? It should be specified in the text. Also the abstract only focuses on P. aeruginosa.
The results are well described but in this section the discussion is poor.
In example, which is the authors’hypothesis regarding the different antimicrobial activity of the extracts against Gram positive and Gram negative bacteria?
The authors only employed type strains, while autochtonous or clinical strains could give different results.
The discussion regarding biofilm inhibition, motility and pigment production in Pseudomonas aeruginosa should be improved, also explaining the reasons why these parameters are important and worthy to be analysed, as the introduction only focuses on quorum sensing. At this purpose the manuscript Rossi et al., Journal of Applied Microbiology (2018), 124(5):1220-1231 could be useful, as well as Hall et al. Toxins (2016), 8(8):26. Please detail the discussion and include the two manuscriptsin your references (if they believe it necessary, obviously the authors could add also other manuscripts useful for the data discussion).
Why did the authors decide to test singularly the effect of selected phenolic components on the Quorum Sensing signalling, if data on this effect were already available in literature (references 17-21)? Which is the added value of these results?
Materials and methods:
L235: why the filtered rhizome extract of B. officinalis did not require a purification step?
Table 3: In the table the results of the analysis seem to be already reported. If those are the parameters investigated at the different hpf, it should be better described in the text and/or in the legend of the table. No reference to the table has been made in the text of paragraph 3.7.
Author Response
Please see the attachment.

This manuscript is a resubmission of an earlier submission. The following is a list of the peer review reports and author responses from that submission.
Round 1
Reviewer 1 Report
The manuscript molecules-736698 demonstrates the development trend of one of the areas of bioactive compounds of natural origin and in my opinion can be published in Molecules.
However, I think that the manuscript can be improved for reading. This will increase interest for a wider range of readers. Here are some suggestions:
- Abstract should show the essence of the article, in my opinion, and not be a summary of what the authors did. It is not clear what the achievements of the authors compared with the known data. These plants are medicinal. What is the fundamental discovery of the authors? What is fundamentally decided, what effect is achieved. Abstract must be corrected.
- The authors do not provide a rationale for the choice of extraction method. Why is ethanol chosen, why is the concentration 80%? How the extraction was carried out is not indicated at all. It is necessary to make corrections so that the data obtained by the authors are useful not only for themselves, since they publish them.
- There is no evidence of the identity of the composition of the alcoholic extract and the solution in DMSO of the dry residue of the extract. When the extract is dried and the residue is dissolved in DMSO, chemical reactions can occur. Authors need to comment on this.
- Conclusion as Abstract should show the essence of the article. It is not clear what the achievements of the authors are compared with the known data. What is fundamentally decided, what effect is achieved. Conclusion must be corrected.
Reviewer 2 Report
The submitted manuscript describes the biological assessment of the extracts of P. maculosa and B. officinalis both for antibacterial/antibiofilm effect and general cytotoxicity. The experimental design is suitable and well reported. My major concerns is that the article is suggesting that these two plant species contain biological active compounds and therefore they could be used on humans for their anti-virulence properties. I think this is a massive leap to make seeing as only the ethanol extract and some pure compounds were tested. Additionally the study has shown that the pure compounds that were identified and individually tested were significantly less active than the extract. A plant is significantly more than just its extract or pure compounds and clearly more should be studied before considering even an extract as a possible therapy. Furthermore, this study does not clearly identify any single compound or compound class which is responsible for activity - this may be due to the fact not all single compounds were tested. At this point I suggest this study is more suitable for a ethnopharmacology or phytochemical journal than Molecules.